# Digital Twins and Enabling Technologies in Museums and Cultural Heritage: An Overview

**DOI:** 10.3390/s23031583

**Published:** 2023-02-01

**Authors:** Wolfram Luther, Nelson Baloian, Daniel Biella, Daniel Sacher

**Affiliations:** 1Department of Computer Science and Applied Cognitive Science, University of Duisburg-Essen, 47057 Duisburg, Germany; 2Department of Computer Science, University of Chile, Av. Blanco Encalada 2120, Santiago 8320000, Chile; 3Center for Information and Media Services, University of Duisburg-Essen, 47057 Duisburg, Germany

**Keywords:** virtual museum, digital twin, emerging technologies, sensor

## Abstract

This paper presents an overview of various types of virtual museums (ViM) as native artifacts or as digital twins (DT) of physical museums (PM). Depending on their mission and features, we discuss various enabling technologies and sensor equipment with their specific requirements and complexities, advantages and drawbacks in relation to each other at all stages of a DT’s life cycle. A DT is a virtual construct and embodies innovative concepts based on emerging technologies (ET) using adequate sensor configurations for (meta-)data import and exchange. Our keyword-based search for articles, conference papers, (chapters from) books and reviews yielded 43 contributions and 43 further important references from Industry 4.0, Tourism and Heritage 4.0. After closer examination, a reference corpus of 40 contributions was evaluated in detail and classified along with their variants of DT—content-, communication-, and collaboration-centric and risk-informed ViMs. Their system features correlate with different application areas (AA), new or improved technologies—mostly still under development—and sensors used. Our proposal suggests a template-based, generative approach to DTs using standardized metadata formats, expert/curator software and customers’/visitors’ engagement. It advocates for stakeholders’ collaboration as part of a comprehensive validation and verification assessment (V&VA) throughout the DT’s entire life cycle.

## 1. Introduction

All areas of society are witnessing the digitization of data, objects and digitalization of services, networked information and communication processes in the areas of industry and administration, production, commerce, finance, logistics and transportation, healthcare and nursing, education and learning, and leisure, sports and tourism—along with all their diverse cultural offerings. This trend is reflected in the introduction of related concepts for digital transformation and has strongly promoted the development and application of enabling and emerging technologies in the aforementioned areas. It has also led to the (sensor-based) collection of data from a wide variety of sources used for the construction of parallel virtual spaces and worlds and their colonization by digital twins that interact with each other. This process requires redistributing tasks and further developing cultural techniques, which contribute considerably to the transformation of the living spaces and habits of humans and animals in all areas of daily life. Digital transformation and technological progress have benefitted the working world through the elimination of simple or physically demanding activities in favor of highly skilled ones and led to breakthrough advances in the sciences, medicine, social life and leisure activities [1,2,3].

### 1.1. The Purpose of the Article

In this paper, we focus on digital transformation in leisure activities, especially all kinds of museums, exhibitions and their DTs. Through a literature search and evaluation, we will provide an analytic overview of the necessary concepts and methods involved in a collaborative effort to produce a cost-effective approach to semi-automatically generated ViMs. This approach consists of digitizing objects and assets depending on their planned features and dimensions and selecting room templates for their exhibition environments, including indoor and outdoor areas, as well as recommended visitor tours to points of interest and interaction options with the items based on standardized museum data and metadata.

From the perspective of the Internet of Things (IoT) (for abbreviations see Table A1 in Appendix A), the ViM implements connected indoor and outdoor exhibition spaces with their artwork and infrastructure, using a network of hardware devices, sensors, middleware and embedded software that collect and exchange data and offer services and facilities to visitors and to other museums as well [4].

Based on a reference corpus resulting from a search for scientific publications in books, journals and conference papers on the keywords *virtual museums*, *digital twins*, *emerging technologies* and *sensor* in leading science databases [1,2,3,4,5,6,7,8,9,10,11,12,13,14,15,16,17,18,19,20,21,22,23,24,25,26,27,28,29,30,31,32,33,34,35,36,37,38,39,40,41,42,43], we classify DTs according to their application areas, relevant technologies and concepts depending on type, focus and features, sensor technology used and adaptation or innovation power throughout their life cycle. Design, realization, operation and evolution are guided by experts, engineers, customers, curators, software engineers and visitors, all of whom define adequate quality criteria (QC) and metrics (QM) for DTs and ViMs V&VA. The results are documented and commented in tables and figures. It can be seen that virtualization and DTs with different features—3D reconstruction, sensory data acquisition, simulation, multimodal interaction, collaboration, and use of artificial intelligence (AI) in virtual networked spaces—have only recently been studied under the label Tourism and Heritage 4.0 analogously to Industry 4.0 with Reference Architectural Model Industry 4.0 (RAMI 4.0), Manufacturing 4.0, Health 4.0, Education 4.0 and their Cyber Physical Systems (CPS) [35].

However, essential aspects have not yet been sufficiently investigated, such as the drawbacks and benefits of the DT concept and the template-based generation of DT with its assets, properties, features and extensions, taking into account international standards and formats for the exchange of data, metadata and assets while applying digital rights management (DRM) and comprehensive lifelong quality management within a concise V&VA [39].

To fully realize the Industry 4.0 vision, it is necessary to integrate multiple relevant feature-oriented DTs (FoDT) of a system according to a specific task description. This requires integrating all available data, information and knowledge related to the system across its entire life cycle [35]. The DT model ensures the evaluation of data from embedded sensors in the simulation of the physical processes and derives predictions for further development, which, after applying machine learning algorithms, may improve the accuracy compared to the theoretical predictions of physical twins (PT) or sensor data alone.

### 1.2. Objects of Investigation, Corpus and Relevant Literature for the Study

It has already been more than 70 years since the development of important elements of virtual exhibitions and museums began. In the article “The virtual museum: An overview of its origins, concepts, and terminology” Schweibenz [44] explains technical basics and mentions important contributors, early ViM realizations and CD-based multimedia products conceived by leading museums as preparation for or supplemental to an onsite visit, which evolved in parallel with the emergence of the World Wide Web, hypertext, 3D-modeling languages, and browsers. However, the concept of digital and physical twins has experienced systematic application only in recent years in the field of cultural heritage and tourism. Industrial processes, modern manufacturing, medicinal practice and their trends towards digitalization and automation associated with Industry 4.0, Manufacturing 4.0, and Health 4.0 and their ETs are juxtaposed with corresponding concepts and developments in the life cycle of a virtual museum depending on its type and goals.

A reason for this might be that this conceptualization tends to focus industry and IT attention on the needs of museums, galleries, gardens, and libraries, and brings to reality the long-demanded development of curatorial tools for planning, designing, generating, and maintaining exhibitions [45].

This kind of software helps curators and experts in planning, presenting and managing content in an adequate virtual environment, as well as in exchanging standardized data and metadata, software libraries, and templates for generating indoor and outdoor museum environments; supporting visitor navigation and interaction with artwork; and making participatory offerings to interested volunteers while implementing privacy and security issues [44,46]. The subject of virtual museums can be very different: art, technology and science, local history, natural history and science, archaeology, ethnography and regional traditions and cultural assets. Furthermore, these subjects can be presented in outdoor museums or museum complexes with various orientations. Concepts, trends, a bundle of ETs and sensor data from a physical environment help to create and operate a ViM as an important part of Heritage and Tourism 4.0. They fulfil complex requirements such as preservation of cultural heritage (CH), sustainable resource management, and smart manufacturing in the reconstruction of assets or realistic process simulation in museums demonstrating historical technologies, such as looms or steam engines.

ETs whose development, practical applications, or both are still more or less unrealized are generally new, including established technologies finding new applications. Used in the context of the (Industrial) IoT or CPS as an integrated, computer-based environment, they must support modeling, sensing, reconstruction, process simulation, control, interfacing, networking, etc. They are manifold and include cross-cutting technologies, such as AI, immersion, simulation, cyber security, cloud computing (CC) and big data used to manage various DT features. This is not a complete list; it includes only the most important technologies. We will discuss such technologies below in greater detail. In our discussion, we do not distinguish between concepts and their technologies.

More general concepts used in ViM and DT, such as Industry 4.0 or Health 4.0; existing orientations, such as preservation of CH, sustainable resource management, and smart manufacturing; digital surgery; and a bundle of enabling technologies that are also used in IoT or CPS-integrated modeling, sensing, big data, reliable computation, control, virtual, augmented or mixed reality (VR, AR, MR) and their interfaces, networking, etc. They include—and this list is by no means exhaustive—the following (numbers in boldface indicated the most cited ET in the reference corpus) [3]:Additive manufacturing, e.g., printing 3D objects from digital models;Artificial intelligence, e.g., vision, natural language processing, deep learning (DL), expert systems, planning & controlling, autonomous vehicles, robotics, etc.; **19**Big data; **7**Cloud computing, cloud systems; **10**Cognitive computing, e.g., to perform humanlike intelligent activities;Communication networks based on various network technologies;Crowdsourcing, collaborative, participative engagement;Cyber security, e.g., block chain, cryptographic protocols; **10**Distributed systems, e.g., wireless sensor networks, cloud or embedded systems;Laser scanning, photogrammetry and drones scanning the assets and ground; **7**Law, governance: big thinking;Immersive technologies (VR, AR, MR) in various application areas and interfaces; **17**Machine learning (ML), machine vision (MV);Neural networks (NN);Product life cycle management;Sensor networks, together with actuators, micro-controllers, smart environments; **7**Simulation; **11**Wearables, e.g., incorporated devices with wireless communications capability;Wireless technologies, e.g., 5/6G, incl. physical devices. **5**

In this review paper, we examine prominent science databases for occurrences of the terms virtual, museum, digital, twin, emerging, technologies, and sensor in journals, book chapters, conference papers and ePublications. Recent searches in Scopus, IEEEXplore, and Google Scholar with the search string “virtual AND museum AND digital AND twin AND emerging AND technologies AND sensor” yield 43 relevant publications [1,2,3,4,5,6,7,8,9,10,11,12,13,14,15,16,17,18,19,20,21,22,23,24,25,26,27,28,29,30,31,32,33,34,35,36,37,38,39,40,41,42,43] from the last 13 years, 34 of which were in Scopus, 3 of which were excluded because the keywords were used in other meanings, such as “Siamese twins” (cf. Figure 1 and Figure A1 in the Appendix A). The authors in [7] proposed dynamic authoring and hybrid recommender methods for accurate AR-based reporting [17] reported on specific human factors associated with and related to technology use in society, both without reference to virtual museums or use of DT concepts; the author of [34] dealt with the microscopic motile fauna of aquatic sediments and twin islands.

Each of the publications was evaluated with respect to the following parameters: Bibliographic data, application area, emerging and other relevant technologies, sensors, taxonomies/classifications/definitions, quality criteria, feature-oriented DT, general DT, time span (TS) and number of references (NR), and future developments.

Due to the global attribution of references in books, these were excluded and for the remaining 33 papers published between 2010 and 2022 (median 2022) the number of references ranged from 5 to 775 (median 100) and covered the period 1915 to 2022 with the median 1993 for the lower bound and 2021 for the upper bound.

Corpus papers found and supplementary articles were analyzed with respect to the following parameters:Type and year of publication with the bibliographic data in the MDPI and ACS style, including the number of references in the respective period;Digital twins and their various types, features and functionalities, benefits and drawbacks, and implications;Virtual museums and their dimensions, relevant standards and V&VA;Quality criteria and metrics for five types of DTs: content-, communication/com-, collaboration/col-, user-centric, and augmented;Emerging and supporting technologies and their importance in the corpus studied;Sensor classes and their technological underpinnings, their prevalence in the publications/DTs/museums studied;Future directions.

The corpus and other relevant references studied have focused on introductory overviews, taxonomies, DTs with different orientations and various features, and augmented DTs in a variety of ways. Before turning to a systematic approach in Section 2 highlighting these papers, we offer a list of these foci: Digital twin modeling [3], digital twin in Industry: State-of-the-art [46], fused twins [4], applications of digital twin across industries [47], digital twin reference model development to prevent operators’ risk in process plants [48], DUET: A framework for building secure and trusted digital twins of smart cities [49], risk-informed digital twins for the built environment [50], urban digital twins [19], digital twin paradigm [36], augmented reality and digital twin system for interaction [37], digital twin augmentation [38], feature-oriented digital twins for life cycle phases and risk-informed DT [39], 3D reconstruction, digital twinning and virtual reality [41], populating the data space for cultural heritage with heritage digital twins [42], and robot DT [43]. A methodological analysis is given in Section 2.

### 1.3. The Structure of the Article

After an introductory paragraph, in Section 1, we introduce an evaluation format similar to index cards, into which we transferred the information from the corpus publications examined in the subsequent sections. A typical publication data sheet is shown in Table 1. Through the concrete input of the search words digital, twin, virtual, museum, emerging, technologies and sensor, we compiled a corpus of publications and complementary literature that allows, in Section 2, the characterization of museums and their digital twins by type and focus, as well as the assembly of systems and their digital twins with different maturity, features and dimensions according to their occurrence in the corpus with the technologies and sensors used. In Section 3, the occurrence of digital twins by publication, year of appearance and features is related to the included technologies and sensors, and conspicuous correlations are derived.

Section 4 is dedicated to the advantages and disadvantages of digital twin technologies, highlighting opportunities for their sustainable further development through the use of cross-cutting tools, reusable templates and AI methods as well as approaches to a thorough V&V management, followed by our conclusions.

## 2. Materials and Methods

In this section, we first introduce concepts of digital twins with reference to reviews, definitions, taxonomies, their scope of services and maturity, architecture, and areas of use, followed by virtual museums focusing on physical twins from Heritage 4.0. International organizations have successfully sought a unified definition of virtual museums and classifications by type and focus, justifying a classification into content-, communication- and collaboration-centric ViMs, which may have further crosscutting features as well. For a list of topics in Section 2, see Figure 2.

### 2.1. DT-Introductory, Application Areas, Definitions, Taxonomies, Standards and Types

Michael Grieves [51] introduced the concept of a DT “as a virtual representation of what has been produced.” This white paper proposed “that the connection between the data about the physical product and the information contained on the virtual product be synchronized.”

Tao et al. [46] present the historical development of the DT concept, theoretical foundations, and the DT life cycle since its introduction in 2007. After a targeted literature search in scientific databases, 74 papers from 2010 to 2018 were selected and DT applications in various fields highlighted. These are the areas of design, production, prognostics, and health management (PHM), concluding with the factory of the future. Enabling technologies for DT are IoT, ML, VR, AR, human–computer interaction (HCI), smart manufacturing, 3D printing, (big) data (analytics) using sensor-based data to diagnose damage size, product location and other failure information. The authors also addressed V&VA issues to a bound model, hardware and algorithm error. A complete DT should include five dimensions: physical, virtual, connection, data, and service. Variants and extensions of the DT concept were not addressed.

Two further papers [47,52] by Singh et al. covered the applications of DTs and their advantages in more than 13 industries/sectors, further mentioning heritage and cites, and the following estimate of DT growth published by GVR [53].

The global digital twin market size was valued at 7.48 billion USD in 2021 and is projected to grow at a compound annual growth rate (CAGR) of 39.1% from 2022 to 2030. Production halts and supply chain disruptions during the COVID-19 pandemic led to the suspension of several other activities along the value chain of industries, such as aerospace, manufacturing, and automotive. This impacted the market during the first half of the pandemic in 2020.

In a structured literature review, Van der Valk et al. [54] examined 233 papers, analyzing the explanations and elements of various DTs and common definitions and proposed a multidimensional taxonomy of DTs by specifying the central features or purposes in eight possible ‘dimensions’ and their characteristics: data link, purpose, conceptual elements-relationship between DT and PT, accuracy-identical or partial DT, interface–human/man machine, synchronization, data input (raw or processed), and time of creation (PT first, DT first or simultaneous).

Several Standards Developing Organizations (SDOs) are developing DT standards such as the International Organization for Standardization (ISO), World Wide Web Consortium or Web of Things (W3C-WoT), the Industrial Internet Consortium (IIC) and the Platform Industry 4.0, etc., as reported in [55]. This contribution highlights the relevance of IoT and DT and the importance of interoperability and standardization for Industry 4.0. A further topic is Digital Twin Definition Language (DTDL), which describes models and interfaces for IoT-oriented digital transformation.

“The ISO 23247 series defines a framework to support the creation of digital twins of observable manufacturing elements including personnel, equipment, materials, manufacturing processes, facilities, environment, products, and supporting documents.” ISO/TC 184/SC 4 [56] contains the following chapters:ISO 23247-1: General principles and requirements for developing digital twins in manufacturing: scope, terms, definitions and abbreviated terms;ISO 23247-2: Reference architecture with functional views;ISO 23247-3: List of basic information attributes for the observable manufacturing elements;ISO 23247-4: Technical requirements for information exchange between entities within the reference architecture.

Various other classification approaches for DTs appear in the literature according to


*Maturity of DT*


Number of functionalities;Scope of their functionalities;Partial, clone (complete), and augmented form, proposed by M. Grieves [51,57].


*Partial DT*


The minimal digital twin typically contains a small number of data sources, i.e., model component or environment sensor data.This level enables quick development of device-to-platform functionalities.A partial digital twin contains enough data sources to create a component’s or asset’s derivative data.


*Clone DT*


At this level, all meaningful and measurable data and metadata sources from the PT are available to generate the DT.This level is applicable when a connected asset is not power- or data-constrained.This level is useful in prototyping and data characterization phases of IoT development.


*Augmented DT*


The augmented DT enhances data and metadata from the connected asset with derivative data, correlated data from federated sources, and/or intelligence (meta-) data resulting from analytics and algorithms.

In the context of a staged V&VA, requirements for the quality of the DT must be set and checked in advance with suitable criteria and metrics. Another four-level-based classification is suggested by [58]:Pre-digital twin together with a virtual system model with emphasis on technology or technical risk mitigation, but without existing PT;DT existing as a virtual system model of PT and required data acquisition;Adaptive DT as a virtual system model with more comprehensive requirements concerning data acquisition, i.e., adaptive user interface, real-time system updating, etc.Intelligent DT with additional learning and decision-making technologies.

Other examples of augmented DTs can be found in the literature.

*Fused DT* stands for the “concept of embedding the DT into the PT through situated analytics” and their interaction, they are visualized in Mixed Reality [4]. Further AI-based features are realized in the cognitive twin.

*Cognitive**digital**twin* (CDT) is a digital representation of a physical system that is augmented with certain cognitive capabilities and supports to execute autonomous activities; comprises a set of semantically interlinked digital models related to different life cycle phases of the physical system including its subsystems and components; and evolves continuously with the physical system across the entire life cycle [35],

More precisely, the CDT definition states the following:(1)Augmented DT including multiple PT, DT and their connection;(2)Cognitive capabilities: performs human-like intelligent activities, follows dynamic optimization strategies;(3)Lifetime management;(4)Autonomy capabilities: reacting to changing requirements and collected sensor data, making decisions;(5)Continuously evolving.

*Historic DT* (HDT) was introduced in [59] and is conceived as a digital replica (DT) of the main historical, geometric, mechanical and structural characteristics of a physical object using IoT, DL, scanning technologies and a data acquisition management system relevant for the maintenance of cultural heritage.

*Urban DT* has the features IoT Server + AR city miniature + AR head-mounted device (HMD) + tabletop interface + 3D city map + geo-localization [19].

*During-time DT (dtDT)* supports communication between the physical environment and the viewer with VR and AR [41].

Redelinghaus [60] describes a layered connection architecture for DTs: A specific architecture is proposed in Layers 1 and 2, constitutes the physical twin and consists of various devices, sensors, and data sources. Layer 3 contains repositories connected to the IoT gateways in Layers 4 and 6. In Layer 4 a data-to-information conversion is conducted, Layer 5 stores historical information about PT and DT and Layer 6 contains simulation and emulation tools, as well as user interfaces.

*Heritage DT* was introduced in [42], which outlines the design of the semantic infrastructure of the data space for cultural heritage and the main features of the proposed *Heritage Digital Twin ontology* and provides some examples of its application. An ontology-based system and a generic paradigm combining a multi-layer, multi-level and multi-perspective approach for architecting and developing DTs is presented in [61].

In contrast, Zhao et al. [62] proceed bottom up and highlight crosscutting-technology-centric DTs with AI, 5G and authentication encryption for (huge) data security.

Different DT realizations may serve a similar crosscutting purpose, for example, dealing with risk issues or robots [43]. A concept of risk-informed partial DT was presented in [63] together with VR. As stated in [39], “few ViMs are intentionally risk-informed and there are no accepted standards for their development”. This paper overviews the features of the ViMs and specifies their risks, the purpose of the corresponding risk sensors and relevant sensor classes. Inspired by [50], it gives an overview of the features of ViMs and specifies their risks detected by sensors from relevant sensor classes. Using traceable quality criteria, their metrics, and targeted user surveys, a formal audit of DTs and ViMs is performed based on given requirements, ground truth data, and related standards from the scientific organizations involved. The proposed verification and validation assessment was developed in joint projects by the Universities of Chile and Duisburg-Essen with the Museo de Arte Contemporaneo (MAC, Santiago de Chile) for the digitization of exhibition spaces using curator tools, with the Salomon Ludwig Steinheim Institute in Essen and the Düsseldorf Memorial to the Victims of Persecution for the reconstruction and presentation of the largely destroyed works of the German–Jewish sculptor Leopold Fleischhacker and during the realization of a virtual Khachkar Museum dedicated to endangered Armenian cross stones in collaboration with the American University of Armenia [39].

In [43] the authors propose a “Cyber–Physical–Social Eco-Society System capable of seamlessly blending the real world with virtual social spaces by intertwining diverse technologies, including real and artificial agents and elements capable of dynamically interacting, reflecting, and influencing each other with the interactions engendered by humans and their behavior.”

The contribution of [36] is devoted to DT design. The authors took a structured approach to the construction of the digital twin and its architecture, functions, components, technologies and life cycle. They justified the introduction of parallel virtual spaces with quality deficiencies, such as poor efficiency and reliability, in the planning, realization and operation of technical systems, buildings and artifacts due to the lack of use of information technology. This development is accelerated by the emergence of concepts and trends such as Industry 4.0 and factors such as big data, cloud computing, cyber physical systems, SMART manufacturing or management in many areas of daily life, where SMART stands for specific measurable achievable reasonable time-bound. After a careful explanation of the approach to the literature review and the textual and visual presentation of the results, the paper examined 30 published definitions of digital twin and classified them according to a hierarchical clustering of terms to C1 (production, product design, product and life cycle), C2 (CPS, digital model, device), C3 (data model machine), C4 (physical and digital model, corresponding behavior, environment), and C5 (control, effective, data, time machine). This results of which are in the following:

“A set of adaptive models that emulate the behavior of a physical system in a virtual system getting real time data to update itself along its life cycle. The digital twin replicates the physical system to predict failures and opportunities for changing, to prescribe real time actions for optimizing and/or mitigating unexpected events, and to observe and evaluate the operating profile system”.

Special mention has been made of formal concepts, technologies and components used for communication, sensors and sensor networks in the physical layer, the network layer, and the computing layer. VR, AR or MR technologies integrated in digital twins help to create interactive and immersive environments enabling direct interactions between the digital twin and the various user groups and their usability expectations.

Finally, important application areas are mentioned: manufacturing, construction, logistics and transportation, healthcare, city management—modeling and simulation of smart cities and the use of DT in all phases of the life cycle with the description of their data, evaluation and optimization of different quality criteria such as accuracy, performance, efficiency and usability.

However, important aspects are missing: end user groups and collaboration issues, technologies in the context of AI and DL used in augmented, fused, cognitive DT as proposed by Zheng [35]: “CDT uses real time data from IoT sensors and other sources to enable learning, reasoning and automatically adjusting for improved decision making”. The DT model provides the evaluation of data from embedded sensors in the simulation of physical processes in PT and derives predictions for further processing, which, after applying algorithms of ML, improve the accuracy compared to the theoretical predictions of PT or sensor data alone.

### 2.2. Virtual Museums, Definitions, Classification, Dimensions, Metadata Formats

For many years, galleries, libraries, archives and museums (GLAM) have played an increasingly important role in education, information, leisure and tourism, preserving cultural heritage and reaching global audiences through motivating more and more citizens to participate in the expansion of their offerings. The EGMUS group [64] was established in 2002. At present, 30 European countries from within and outside the European Union are represented in this group. The main objective of EGMUS is the collection and publication of comparable statistical data. Available data from national museum statistics and surveys are compiled, updated, and stored in the Abridged List of Key Museum Indicators (ALOKMI).

If we add up the responses from 31 European countries in the various categories for the last reporting year, we arrive at approximately 20,000 museums with more than 500 million visitors per annum.

Since 2010, the organization OpenGLAM [65] has been coordinating various ways to access (virtual) cultural heritage sites and supporting related projects. In 2018, members of the Creative Commons, the Wikimedia Foundation and Open Knowledge started plans to revitalize the initiative. By defining individual- and GLAM-generated intellectual property and granting access to Open GLAM, the organization, together with museum associations around the world, strives for copyright consensus, knowledge dissemination and the preservation of tangible and intangible cultural heritage.

According to the definition given by the International Council of Museums (ICOM), the German museum association [66] presents in this annual publication a museum classification as a three-dimensional vector (number of museums, number of temporary exhibitions, number of visitors in millions) for the following museum types: *(regional) history/ethnography/archaeology museums* (2814, 4101, 14.49); *art museums (museums of sculpture, picture galleries, museums of photography and cinema, and museums of architecture, including art exhibition galleries permanently maintained by libraries and archives centers)* (674, 1764, 18.63); *castles with inventory* (274, 241, 13.13); *natural history and science museums dealing with biology, geology, botany, zoology, paleontology and ecology* (297, 327, 7.33); *science and technology museums such as astronomy, mathematics, physics, chemistry, medical science, construction and building industries, manufactured objects, etc.; also included in this category are planetaria and science centers* (779, 770, 18.16); *archaeology and history museums* (468, 492, 20.81); *overarching museums with various collections* (29, 122, 2.41); *specialized museums* (937, 880, 10.93); *museum complexes with various museums* (86, 245, 4.48). One hundred forty outdoor museums were not listed independently, but are classified within these nine categories. Of the museums included, 53% are operated under public administration, 43% are operated privately and 4% have a private and a public governing board. Further museum type definitions, collection related tasks and other terms used in museum statistics are highlighted in [45].

ISO 18461:2016 [67] provides state-of-the-art examples for ViM, a definition of the term virtual museum and an initial classification system reflecting eight ViM dimensions: content, communication, duration (time of use), format (distribution), interaction technology, level of immersion, scope, and sustainability.

In analogy to our contribution to the classification of ViMs and from our experience with the creation of three major virtual museums between 2010 and 2020 with partners from Armenia, Germany and Chile, we propose for DTs three main dimensions: content/product-, communication-, and collaboration-centric ViMMs (cf. [39,68,69]). For each of the three additional dimensions/features (cf. Figure 3), we suggest further overarching dimensions such as AI and ML, and privacy and security (including those based on sensor technologies, such as the risk-, or bio-informed DT). A DT/ViM realizing bidirectional communication between asset and customer/visitor or user participation is user-centric; including further AI-based features makes it a CDT.

In order to further develop the coexistence of physical museums and ViMs, Shehade and Stylianou [70] “explore the practices, experiences, and perceptions of museum professionals related to the use of VR technology in museums, as well as the advantages and challenges of such technologies, by evaluating an in-depth analysis” and state that “VR and other emerging technologies create a fundamental shift, necessitating in some cases a reinvention of traditional concepts on what the museum experience entails”.

Damala [71] champions the MUSETECH Model. The framework evaluates museum technology before and after its introduction into a museum setting from three different perspectives—the curator’s, the visitors’ and the institution’s.

Schweibenz [44] highlights the fundamentals of ViM and its relationship to PM through evolving definitions of professional organizations—the International Council of Museums (ICOM), V-MUST [72], ViMM, etc.—and their representatives from 1994 to 2019 in light of technological and societal developments.

The 386-word ViMM definition first cites the now outdated ICOM museum definition Article 3, Statutes, 2007. It then references the PM, the relation between DT and PT and the classification of DTs (clone, augmented, feature-oriented, content-, communication-, collaboration-centric and subordinate dimensions) and stakeholders with their various tasks and opportunities over the entire life cycle of the ViM. An augmented ViM features unlimited ubiquitous access to all resources and services—even physical or born digital assets—from the entire museum network and from sites in situ otherwise inaccessible.

In addition, the ViM provides new opportunities for the public in ways not previously viable through the modification, experimentation and combination of collections based on the digital iteration of the analog original. Users actively participate in novel ways such as through enhanced ‘presence’, immersive experiences, multidirectional communication, and interactive storytelling. Finally, it addresses overarching aspects, such as “technical innovation, guidelines of validation, DRM novel curatorial scenarios“, as well as enhanced user offerings for pre- and post-visit preparation.

As reported in [39] and in addition to PM definitions published by the International Council of Museums, “the website V-must.net strives to clarify the term virtual museum, providing state-of-the-art examples for ViM and an initial ViM classification system reflecting a dimension of a given ViM. According to this classification, possible dimensions are content, communication, duration (time of use), format (distribution), interaction technology, level of immersion, scope, and sustainability”.

**Figure 3 sensors-23-01583-f003:**
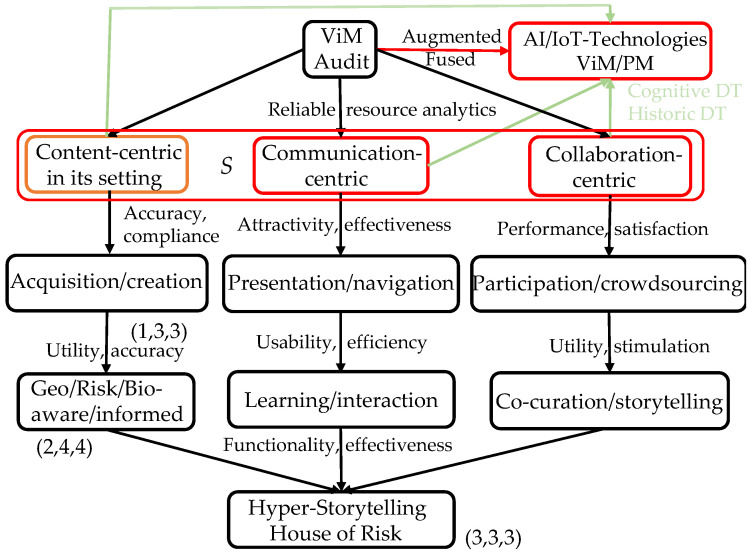
ViMs and their dimensions, DTs and their variants [73].

The category content-centric ViM (focusing on artwork and its setting) is superordinate to the other categories and of particular importance for virtual museums since here—in addition to the content—its surroundings, rooms, and a logical floor plan with navigation elements must be designed and implemented. This leads, especially when real existing objects are digitized, to large data sets and their metadata.

In this context, the mathematical dimension concept for a Euclidean vector space with three space dimensions and one time dimension is used to allow event-based models, and further parameters for the surface shape and color, lighting, appearance and history of the artifact. Another conceptual category, communication-centric ViM, is about knowledge transfer, learning, linking to related objects, forms of presentation, navigation and interaction, standardized data and metadata exchange and the use of appropriate formats, interfaces and output devices.

The collaboration-centric ViM includes appropriate web platforms with shared workspaces and participatory approaches, such as crowdsourcing and co-curation. Hyper-storytelling based on georeferenced archaeological artifacts belongs to the last two categories. Extensions of the DT/ViM concept include AI, IoT and other overarching technologies, as well as cognitive capabilities and autonomous DT enhancements.

The concepts realized in these categories are often called ViM dimensions, with a definition of dimension taken from graph theory: the minimum number of vertices of an undirected graph *G* in a subset *S* of vertices such that all other vertices are uniquely determined by their distances to the vertices in *S*. Here, for the definition of the set *S*, three child nodes of the root were selected; then further nodes may exist on a path down to a leaf, which is the node with the largest distance corresponding to the complete implementation of the category with the subcategories or types. Along these paths, the ViM dimensions are then evaluated (cf. Figure 3) with respect to the annotated quality criteria. Rather than a graph visualization, a regular *n*-sided polygon could be used. If the feature associated with a node from *S* appears to be too general or comprehensive, it can be split into multiple nodes to increase the graph dimension [73]. In the case of augmented DT, a fourth vector component must be added that counts the overarching features.

For a smooth transformation from PT to DT, it is especially important to use appropriate metadata exchange formats. At this time, there are approximately 100 CH metadata formats available.

The authors of [74,75] investigate artifact-oriented ontologies and metadata schemes for the acquisition process to provide details about the 3D capturing, which can be combined with already existing ontologies, e.g., CIDOC CRM2 [76] and collect current applications by CH data providers of technological solutions in use for making structured data (or metadata, in the CH context) available for re-use in the internet.

Below, we list important groups in the context of (virtual) museums with examples (cf. [39]):Strictly ViM formats: Virtual Museum and Cultural Object Exchange Format (ViM-COX) [45], Europeana Data Model [77], and TourML [78], an XML schema that describes the structure, content and experience of a ViM mobile tour (navigation supporting the feature communication);Museum standard LIDO (lightweight information describing objects) is the successor of the metadata exchange format museumdat and was inspired by CDWA Lite and SPECTRUM [79]; LIDO is CIDOC CRM (Comité International pour la Documentation−Conceptual Reference Model) compliant and can be used to document properties of all kinds of cultural heritage;Architectural standards: CAD, IFC/BIM, CityGML/ IndoorGML developed by Open Geospatial Consortium (OGC) (addressing the feature physical structure, content [80]);Educational metadata standard: IEEE Learning Object Model (addressing the feature learning) [81].

A current example of an interchange and harvesting format is the XML schema LIDO [82], created in international cooperation and used to document properties of all kinds of cultural heritage (addressing the feature exchange). The format ViMCOX mentioned above was developed as an application profile of LIDO in order to provide a semantic structure for exhibits and complete museums [83]. It combines community contributions to administrative and descriptive metadata with technical and usage metadata provided by the institution operating the museum.

## 3. Results on DT Types, Emerging Technologies, and Sensor Classes

In this section, we review the occurrence of the keywords DT, ViM, ET, and sensor, in our corpus and explain their functions, and relationships to each other in the DT and ViM life cycle, respectively. Important relations are highlighted, e.g., communication-centric DT related to VR, MR, and XR technologies and multimodal sensing, as well as content-centric DT, 3D reconstruction and printing, TLS and photogrammetry (cf. Figure 4).

### 3.1. Overview

In the following sections, we will first investigate which types of DTs and ViMs, as well as which specific emerging technologies and sensor classes appear in the corpus studied (cf. Table 2).

In conclusion, the most often mentioned DTs of the communication-centric type (incl. HMI, 22) are correlated with AR and VR (17), as well as smartphone built-in sensor technology and sensors to emulate the five human senses ahead of the content-centric DTS (13) with 3D reconstruction and printing (9) and the associated sensors for the identification and digitization of assets (24).

### 3.2. (Virtual) Museums in the Reference Corpus

Below are listed the types of museums addressed in the corpus under review. Thus, all relevant types are represented.

Archaeological museum [19];Architectural heritage museum [40,41];Art and history museum [30];City museum [20,38]—virtual replicas of a city or digital representation of urban spaces as a communication-centric DT with interactive technology [25];Computer history museum-online exhibition [15];Cultural heritage assets, historical noble palaces, national art museums [33,42];Learning-oriented ViM [23];Outdoor museum [38,39];ViM of Robotics, web-based museum [43];Technology museum [29], The Bocholt Textile Factory−LWL Industriemuseum [39];Tourism, city museum [4];Virtual museums with various features [12,39,42];Zoological museum [34].

With regard to the use of IoT, Lerario [25] points to new models for communicating cultural content, which are being adopted by more and more virtual museums, including those cited mentioned above, e.g., [39]. Interactive interfaces, guided tours with selectable points of interest and additional offers of information on the exhibits taking personal interests into account play an important role. Depending on the age of the visitors, different types of terminals or mobile devices can also be available that use video or game-like offerings, or cave-like technology with immersion. In this context, user experience evaluations help to improve the design of interaction with the items that allow their deconstruction and reconstruction, users can prepare and follow up their visits and share their impressions with other visitors [83]. Exhibition design also increasingly involves volunteers as co-curators, who take on important sub-tasks in the selection and design of optional digital offerings. On the part of the curators, visitors’ museum visits, dwell times and activities are recorded and evaluated anonymously. Sensors and beacons for indoor localization and user tracking play an important role here. Many of these developments towards an individual visitor experience have also found their way into physical museums. For example, robots take over advisory tasks, and sensors monitor compliance with the requirements for the environment and presentation of the exhibits as their integral components.

The publication [43] points to the current generation of museum visitors, who are influenced by their exposure to modern technology such as social media, smartphones, IoT, smart devices, and visual games, by providing a unique experience of exploring and interacting with real and virtual worlds simultaneously.

### 3.3. Sensors in Our Corpus

Sensors are used for perceptual data acquisition by people, humanoid robots or their virtualizations. They may acquire data from various social environments, including work and cultural or leisure activities. They may also acquire data from digital systems and their components or processes, such as data for the reconstruction of content or digitizing and operating systems for controlling and monitoring their functions.

Sensors for *measuring or recording owner’s data*, detecting professional and leisure activities [2,22] and scanning the environment, e.g., city life [20], have found their way into many personal technical devices used in everyday life. Beacons for indoor positioning in an interactive IoT-based museum, use of sensors and actuators to minimize the invasive character of conservation actions or influences from outside (such as sunlight, moisture and water level), sensor networks for asset multipurpose measurement and location-based services are highlighted in [25].

From 16 papers, we have derived a short list of *smart phone and smart watch built-in sensors*: GPS, gyroscope, accelerometers, inclinometers and video cameras [21,33] are used in IoT [10,15]; digital identification RFID [25,26,36], indoor positioning and visitors’ motion tracking are used in DTs [18,30,37,40] and in the supervision of exhibits/products, and environmental sensing [25,31,32,39,42]. Motion tracking sensors are needed for monitoring device movement, such as tilt, shake and rotation (velocity). It is also possible to measure position, altitude, acceleration and proximity or distance to an item as needed for navigation and interaction.

*Environmental data* are relevant for sensing in risk-/bio-informed DTs [3,39] and include temperature, humidity, ambient light, barometric pressure, magnetometric, seismometric and hydrometric information to detect weather anomalies, as well as fire, smoke, earthquake and flood [4,11,18].

*(Wearable) physiological sensors* were used in eight communication-centric DTs for health monitoring [33]. Such sensors can collect blood pressure and pulse; skin sensors [16] identify the wearer’s current glucose level. A further sensor group uses digital technologies to augment or emulate the five human senses touch, sight, hearing, smell and taste and are used in VR and XR applications: multimodal sensors [12], smart glasses [5,6,40], optical sensors [13,27], gloves (with optical sensors) [13], hand/body trackers, helmets, external sensors [6] and eye trackers [12].

*Laser or drone scanning, remote sensing*, LiDAR (light detection and ranging), 3D-imaging sensors, close- and midrange terrestrial laser scanning and structured light and multi-beam sensing [14,24,28,29,38] are used for image capturing, area mapping and identifying objects and their properties in five content-centric DTs.

Remote sensing is done by camera, laser, etc. from an aircraft or satellite [4] or from drones [10,15,43]. It measures emitted or reflected radiation (in agriculture [2]), realizes inertial localization [5,19] or generates point clouds [14,28]. For 3D reconstruction in content-centric DTs, objects or artwork are scanned in their surroundings using camera [1], CCD (charge-coupled device) [9], laser [8,29,42], photogrammetry, triangulation in point clouds or other methods [9,14,24,38,40]. Such sensing for reconstruction occurs in seventeen content-centric-DTs.

This classification uses the following keywords in the corpus (and specifies the first and following references):Cameras [1];Activity recognition [2];Sensing in risk-informed DTs to diagnose damage size and location and other failure information [3,39];Remote sensing [4], in agriculture [2] (measuring emitted or reflected radiation by satellite or aircraft);Sensors collect data on temperature, humidity, noise and people’s movement [4,18] to diagnose damage size and location and other failure information [4];Satellite sensors gather data for inertial localization [5], sensing for point clouds [14,28];Extended reality (XR) smart glasses [5,6,40];3D interaction techniques: glasses, gloves, hand/body trackers, helmet and external sensors [6], eye trackers [12,18];Scanning: Laser [8,29,42];Photogrammetry [9,14,24,38,40], CCD [9], drone sensing [10,15,43];Embedded IoT-based sensors [10];Sensors for ecosystems [11];Multimodal sensory perception that handles speech, dialogue flow, gestures, shock, pressure [12];Optical sensors [13,27], gloves (with optical sensors) [13];TLS: 3D model reconstruction, object recognition, deformation measurement, quality assessment, progress tracking, ground penetrating radar [14];Drones that scan the ground, autonomous vehicles, wearable sensors, sensor networks for recording physical activities, multiple sensors connected through the network of IoT [15];Biosensors, embedded sensors in smart wearable devices, wearable sensors for healthcare and human movement monitoring, taxonomy of sensors and data collected via wearable devices [16];Motion tracking [18,30,40];Smart sensors, human sensing, 360° camera, geo-location [19], IoT sensors [35];Sensing data about city life [20];Built-in smartphone sensors, sensing for fostering playful interaction through 360° VR technology [21];Sensors for gamification [22];Wireless sensor network for monitoring air quality [23];Laser or drone scanning, remote sensing, 3D imaging sensors, multi-beam sensors, close-range and midrange terrestrial laser scanning [24,28,29];Structured light sensor [24];Environmental sensing [25,31,32,39,42];RFID technology [25,26,29,36];Proximity sensors [30,39];Inbuilt mobile phone sensors [33];Physiological sensors [33];Ultrasonic distance sensor, accelerometer and gyroscopic sensor [37];Sensory enrichment for understanding and naturally interacting with space [41].

### 3.4. Clustering Emerging Technologies

First, we classify entries in the corpus according to their main application areas and group papers of the same type to cluster them by emerging technologies and concepts.

[1]: Extended reality applications in Industry 4.0 machinery → [1,37];

[2]: Artificial intelligence and machine learning technologies in research, society and industry, costs and benefits of digital (image) forensic investigation → [2,32];

[3]: Smart manufacturing → [10]: Industry 4.0 model factory; [15]: Digital economy, digital transformation in manufacturing, reconfigurable manufacturing systems; [26]: Assembly systems in Industry 4.0; [27]: Digital manufacturing Industry 4.0: Paradoxes and conflicts; [35]: smart manufacturing: RAMI 4.0; [36]: Predictive/smart manufacturing as content-centric DTs → [3,10,15,26,27,35,36];

[4]: A review of access to digital twins in situ in smart cities (smart cities, BIM, CIM, VR, geo-visualization) → [11]: Ecosystem-Based adaptation: Approaches to sustainable management of resources in smart cities; [19]: Virtual urban digital twin system augmenting virtual and real geo-spatial data; [20]: Digital participatory planning in cities; [25]: Sustainable conservation of cultural resources in Italian heritage cities; [31]: Spatiotemporal event visualization, Human–computer affective multimedia interaction, event visualization on Google Earth system, geo-visualization; [33]: System for real-time multimodal analysis of nonverbal affective social interaction in user-centric media → [4,11,19,20,25,31,33];

[5]: Impact of extended reality in Health 4.0, HCI on patient surgery → [6]: Virtual reality applications in cardiology; [21]: Augmenting emerging hospitality services → [5,6,21];

[8]: Digital 3D technologies for humanities research and education → [13]: VR education support System; [22]: Mobile learning as the key to higher education innovation; [23]: Determining significant classification factors for senior learning; [41]: 3D reconstruction, digital twinning, and virtual reality: Architectural heritage applications → [8,13,22,23,41];

[9]: Photogrammetric 3D reconstruction—terrestrial laser scanning (TLS) in architecture, engineering and construction → [14]: Application of terrestrial laser scanning in the architecture, engineering and construction (AEC) industry; [24]: A history of laser scanning, part 2: The later phase of industrial and heritage applications; [28]: Geometric digital twinning of infrastructure. Geometric DT modeling process, Scene understanding, sensor data from buildings; [38]: Drone-based AI and 3D reconstruction/photogrammetry for digital twin augmentation → [9,14,24,28,38];

[12]: Metaverse: Taxonomy, components, applications, and open challenges → [39]: Assessing the use of emerging technologies in feature-oriented digital twins of physical museums via reliable virtual museum analytics; [40]: Protocols for the graphic and constructive diffusion of digital twins of the architectural heritage that guarantee universal accessibility through AR and VR; [41]: Digital twinning, and virtual reality: Architectural heritage applications; [42]: Populating the data space for cultural heritage with heritage digital twins; [43]: A prototype that fuses virtual reality, robots, and social networks to create a new cyber–physical–social eco-society system for cultural heritage → [12,39,40,41,42,43];

[16]: A survey on wearable technology: History, state-of-the-art and current challenges: Disability and haptic mobile media →[18]: The state-of-the-art of spatial interfaces for 3D visualization; [30]: Disability and haptic mobile media. New media and society → [16,18,30].

Next, we sort the papers from group 1 into group 3, and from group 2 into group 8 and assign the corpus to the remaining seven application areas (cf. Table 3).

By doing so, we found that the most frequently mentioned communication-centric DTs (incl. HMI, with 22 mentions) correlate with AR and VR (17), as well as with built-in sensor technology and sensors to emulate the five human senses ahead of the content-centric DTs (13) with 3D reconstruction and 3D printing (9) and the associated sensors for the identification and digitization of an asset (24).

### 3.5. Quality Criteria and Metrics

In [39,84], we advocate a broad V&VA on ViMs. Interested parties should:Strictly follow ViM formats: Virtual museum and cultural object exchange format;Define requirements, QC, and QM for the outcome of the process or task under consideration and their analysis in the early stages of their life cycle;Assess the suitability of the concepts and tools (e.g., for the design, construction, operation, and evolution of the ViM in its life cycle), balancing costs and risks, and making appropriate recommendations.

Several realizations of ViMs and virtual learning labs at UDE and partner universities served as proof of concept. Apart from life cycle management, these projects showed how to involve people in collaborative and crowdsourcing activities.

The overall goal of reliable ViM analytics is to provide viable validation for a range of museum areas with their corresponding QC and QM. Evaluation can be performed through empirical assessment, for example, collected in a collaborative project in which museum stakeholders, curators, visitors (of different experience levels), and domain experts assess the subjective reliability of a ViM domain. In [45] (Appendix A), a questionnaire for curators on this phase of exhibition design development is presented. It consists of 56 items “to gather relevant information about the collections, digitization and metadata techniques and interoperability standards used by the corresponding institution as well as to determine possible use case scenarios for virtual museums and relevant feature sets for future curator tool implementations”. Actual information on the status of the digitization of exhibitions can be found in [69].

In addition, formal descriptions and models of museum entities, benchmarking and best practice guidelines, as well as formal validation (based on standards published by professional societies), provide an objective level for empirical reliability analysis and can be used to validate the system against requirements. Important QCs are accuracy, performance, and utility.

Accuracy in this context, for the data used or provided, means the correctness of their interpretation by the selected data types, fidelity of generation, structural or content integrity, and consistency (i.e., preservation of their logical coherence during processing).

Fidelity refers to the degree of similarity between museum objects and their virtual instances in terms of appearance, size, color and shape or form. Evaluation of this criterion requires a reference or guaranteed error bounds, i.e., choice of appropriate scanning and reconstruction platform and calibration.

Performance is a general term that evaluates the capabilities expected of and services provided by stakeholders and includes efficiency and effectiveness. Efficiency assesses resource use in terms of time or cost (e.g., ratio of (average) time used to total time spent completing tasks); effectiveness assesses task completion.

Fitness/utility assesses the number of resources and their adequacy for stakeholder needs.

Important QC and QM found in the corpus are assigned to the five central DT dimensions in Table 4.

## 4. Discussion

### 4.1. Heritage DTs and Virtual Museums, Augmented DTs—Benefits and Drawbacks

This section deals with the advantages and problems related to the topics of the paper, DT and its emerging technologies, virtual museums, sensors and applications. Typical drawbacks of the DT are its complex design and generation processes depending on PT and the required features. Such drawbacks can lead to a number of quite different partial DTs to save costs. Stakeholders must clarify which features the DT can do without and which additional ones should be implemented.

It must not be overlooked that due to the great variety of museums, gardens, exhibitions and their diverse orientations and offerings to the visitors, very different DTs have to be generated and operated. Therefore, a data- and metadata-related generative approach based on internationally accepted formats is urgently needed. Such an approach must allow for DTs to be networked within a large family of comparable solutions using the automatic generation of exhibits and their surroundings, e.g., with smartphones, photogrammetric methods or ready-made templates.

Cost factors such as the complex generation of many different DTs, the limited scope compared to dissimilar PTs and expensive software maintenance must be weighed against the unique advantages in connecting the physical and virtual spaces of industrial systems. Accurate information for the digital twin (DT) generation of existing assets is provided in [29].

On the other hand, DTs have attracted enormous attention from both industry and academia since their appearance due to their advantages for improving data interoperability and constructing various capabilities. The benefits include modeling a range from preventive maintenance to operational efficiencies and cost savings [36].

Only in a few contributions are concrete consideration and comparisons of the advantages and disadvantages of the object of this investigation found.

Due to its advantages, VR technology is widely used in research, industrial and educational areas. Modern forms of education have some disadvantages, such as the high costs of creating VR environments or purchasing specialized software, as well as the time required to prepare exercise scenarios and tasks [5,6,40].

VR applications for heart, valve replacement and preoperative simulation of a patient, for example, offer the opportunity to solve complex medical problems and fulfill associated requirements. They also promote better communication between health professionals and the patient before performing actual heart surgery. Benefits of this technology also include increased performance, productivity and error reduction.

Advantages of user participation and collaboration are reported in [12,16,17,20,24]. Further topics addressed are the following:Limitations inherent in the current state of AI technologies [2];Combining different interaction paradigms to leverage their inherent benefits and mitigate their limitations has been the focus of multiple research projects [13]; the survey on spatial interfaces highlights the benefits of each interaction paradigm that are most applicable to addressing the challenges of 3D spatial visualization [18];Comparison of localization technologies for wearables, advantages/disadvantages [16] (Table 4) [19];Computer-aided manufacturing and automation software packages, including systems that take advantage of generative technology in design techniques; detailed interests and limitations of 3D printing in relation to the design of an object are addressed in [27];Benefits and limitations of using 3D technologies [8]. Advantages and disadvantages are summarized in various tables dealing with scene understanding, primitive shape detection methods, model generation, and digital 3D representation: comparison of methods for 3D fitting (boundary representation and constructive solid geometry examples) and of common nonlinear optimization techniques [29];The main advantages of TLS over traditional measurement technologies—architecture, engineering and construction—are discussed in [14]. Drone-based augmented DT with reusable and customizable components: extensive evaluation of a proof of concept is conducted for 3D reconstruction and applications of AI for defect detection. The authors of [38] evaluate the performance of three different 3D-scanning technologies with photogrammetry (Pix4D), stereovision (Dot3D/Navisworks) and 3D LiDAR (geoSLAM/Navisworks). Handheld and drone-based scanning with respect to distance error using stereovision/2D LiDAR approaches are compared to manual measurements [38];Various quality criteria have been acquired through reliable ViMs implementing (features of) PMs, which has the additional advantage of possible risk assessment. Reliable ViM analytics should be part of an enhanced V&V management within a workflow for designing, modeling, implementing and validating various installations and tasks in modern feature-oriented virtual GLAMs. In this way, reliable feature-oriented DTs of PMs can be obtained [39];Tables highlight various platforms according to their capabilities with respect to the generation of a DT, photogrammetry, and AR/VR from DTs [40].

### 4.2. Further Work Addressed in Our Corpus

The term digital twin for a virtual museum or virtual exhibition has been used only recently although similar approaches to virtual exhibitions have existed for many years.

Substantial virtual tours in large museums are only available in a few cases. For example, the Albert Hall in London offers a visit using Flash, which is no longer supported. Research should be done to see which museums offer DTs or a partial virtual version of the museum.

The contributions 73] and [85] describe tools for the creation of a virtual exhibition by curators with a room template and 3D authoring techniques based on metadata including visitor–artwork interactions and user evaluation.

Retrospective virtual exhibition visits, a virtual repository of all exhibitions over a certain period of time and access to museum stacks via a ViM are all overdue.

Further work should address taxonomy for risk/AI/bio/geo-informed (fused) DT applications and provide a generic development framework. Further suggestions for future work are found in the corpus:Research oriented towards overcoming the limitations of AI and ML usage are in Table 3, [2];Fused DTs could also be expanded to communicate city planning initiatives to the public [4];Focusing on immersion, the sensory construct requires further investigation to understand its effects on fostering playful interaction through 360° VR technologies [21].

A more complete account of future technical and methodical issues found in the corpus is provided in Table 5.

Issues concern (a) interaction and devices, (b) thematic and methodological issues, (c) new approaches in application areas, and (d) new DT-related insights gained through the application of special techniques or methodologies and attempts to strive for the advantages and overcome the disadvantages.

Our discussion has made clear that modern software technologies relying on a generative approach; 3D asset digitization and 3D reconstruction; room, furniture and landscape templates for indoor and outdoor museums; and exhibition designs described with standardized metadata should be used to generate content-centric DTs in a more cost-efficient way. Such an approach could even be extended by interaction and collaboration design [18,45,68].

The design, realization and operation of communication- and collaboration-centric ViMs should take the following into account: different behaviors of visitors and experts depending on the type, suitable local and temporal organization of access to the exhibited works in their sequence, opportunities of common tour planning and interaction with the exhibited objects, additional information offers and ways of reporting on their impressions [73].

Virtual reality, artificial intelligence and the use of robots should be integrated into the design, generation, operation and evolution of ViMs. For example, convolutional neural networks can be exploited for 3D model reconstruction, asset classification and cataloguing, data retrieval and matching, and visitor support with learning resources. Chatbots can automate communication with visitors and support learning activities. Robots can serve as museum guides in a variety of capacities and interact with visitors, answering questions and telling stories. An important step towards reliable risk communication in museum monitoring is the implementation of a pool of relevant sensors with their interfaces providing data and metadata that should be forwarded to the evaluation logic [39]. Such a pool would support uncertainty quantification and decision making within the computational platform for risk assessment as described in [50].

### 4.3. Implications

The DT models, technologies and example applications described will contribute to a comprehensive discourse on the practical applicability of this approach based on the proposed V&VA for all features and dimensions of virtual museums. This study raises awareness of the various actors playing a role in the development and use of technology to enable museums to reach more people and, at the same time, help researchers identify new research directions. This primarily concerns the stakeholders’ understanding of their part in the life cycle of a ViM, which is rich and complex. Curators should realize that virtualization is a viable way to make the many unexploited treasures in the stacks of major museums and in private collections accessible to a wider public and the exhibited artworks accessible from anywhere while, at the same time, protecting them from destruction. To do this, stakeholders must work closely with software engineers, museum experts, visitor groups and funders to strengthen exchanges among museums, to standardize curatorial tools and visitor support and to expand participatory offerings.

Software engineers, in conjunction with cognitive scientists, help to develop curatorial tools with modern instruments, software generators or templates from software repositories. These tools not only help to digitize and make static exhibitions accessible with scanners or smartphones, but also increasingly use participatory approaches such as crowdsourcing and co-curation to make interactive offers to various visitor groups according to age and interest. Such approaches range from 2D representations in online catalogs to the use of multimodal user interfaces and immersion, which are known to have a great impact on learning processes [37]. Only in this way can interest be generated and added educational value be achieved. To be as effective as possible, software engineers must understand that expanding physical museums through new offerings such as walkthroughs of past times and cultures or of installations by modern artists that use light, sound and movement to engage all the senses is indispensable.

Standards for the description and protection of artifacts and their realistic environments with physical effects extended by virtualized sensor technology to realize bio/risk-informed ViMs must be established and agreed upon. Only in this way will the prerequisites for a successful collaborative implementation be met.

Finally, those responsible in society and politics must equip the museums in such a way—and this also applies to appropriate employee training based on relevant curricula and the promotion of museum competence centers—that they can be successful. Success includes achieving the goals set out in the detailed innovation programs of the national and international museum associations, such as increasing the number of visits [43]. To this end, manifold experiences with simulations of complex technical systems and with project planning and monitoring can be brought in from industry. In addition to institutional sponsors, private donors are also sought who recognize the new possibilities for presenting and preserving cultural heritage and are willing to support them.

In the following references from the corpus we examined, various implications for ViMs and DTs and their enabling technologies are addressed:

Schweibenz [44] mentions sociotechnical implications of the ViM. Concerning ET, [8] provides a detailed discussion of the implications of 3D technology relevant to ViMs. Lerario [25] categorizes “the multifaceted implications of mutual relationship between the IoT and the built cultural heritage” and mentions that “the People, Activities, Context, and Technologies model from the user experience domain will be useful to identify main concern areas that can be affected by them.” To et al. [38] provide some theoretical and empirical implications for researchers as well as practitioners in drone-based AI and 3D reconstruction for DT augmentation. More generally, Bevilacqua [48] describes practical implications of the proposed DT reference model and its use “to detect, identify and develop corrective actions that can affect the safety of operators, and further parameters of the model.”

## 5. Conclusions

Although introduced over 70 years ago, virtual exhibitions parallel to those in physical museums are still not the norm. The increasing threat to the world’s cultural heritage from wars and improper use urgently requires the digitization and presentation of all collections according to uniform norms and standards, respecting digital property rights and with the cooperation of all groups involved. Even if Heritage 4.0 is a relatively small area in terms of projects and publications, the diversity of museum types and exhibition forms—from art museums to open-air exhibitions of historical buildings or technical equipment from the beginnings of industrialization—makes it necessary to call in experts from the respective disciplines to reconstruct, digitize and implement the exhibits and processes, to describe their tasks in concrete terms and to support them in the implementation. Only through a cooperative effort among all the stakeholders involved, including visitors, can curators design these new virtual exhibitions, involve the target audiences and align the project with the orientation of their respective museums. For this purpose, the new frameworks with digital twins and their multiple manifestations and modern taxonomies are very helpful.

In addition to presenting the theoretical basics and required technologies for DT and ViM, we have offered advice for practical implementation using a manageable and generally understandable corpus of relevant literature. Based on recommendations from professional organizations, we have provided stakeholders with references to taxonomies, standards and persistent and current developments regarding DTs in the field of cultural heritage in a comprehensible manner. We have addressed stakeholders’ needs through detailed investigations of relevant applications from the CH area; various definitions of DTs and ViMs; their focus, features and dimensions in relation to enabling technologies and sensory constructs; and their advantages and disadvantages, as well as outstanding work and open problems. Based on concrete referencing in our corpus, in Section 2, Section 3 and Section 4, we highlighted important issues in the four phases of a DT’s life cycle design, realization, operation and innovation. Artwork and physical objects are digitized using various scanning approaches, and physical models and processes are digitalized including path planning or interaction design and description of outdoor areas. The use of robots and AI in museums was also suggested.

Finally, we would like to point out that the findings and conclusions of this study concerning the life cycle of a DT essentially relate and are limited to the field of Heritage 4.0 and ViMs of all categories. While they provide an image of essential areas of human society, these fields focus more on the past than on the future and are nowhere near as complex as the modern issues and solution approaches in the subareas of Industry 4.0. In this respect, methodologies and technologies from computer science, (software) engineering, and architecture are recommended and demonstrated in use cases primarily in order to structure the processes and make them more cost-efficient.

## Figures and Tables

**Figure 1 sensors-23-01583-f001:**
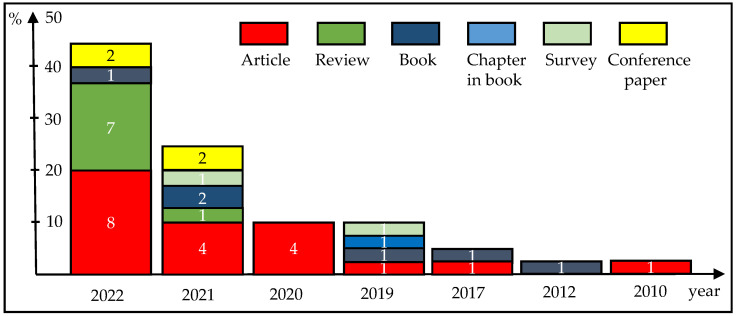
Type, number and year of publication.

**Figure 2 sensors-23-01583-f002:**
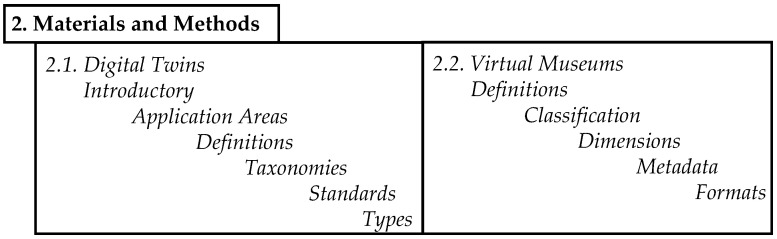
Section 2, subsections and topics.

**Figure 4 sensors-23-01583-f004:**
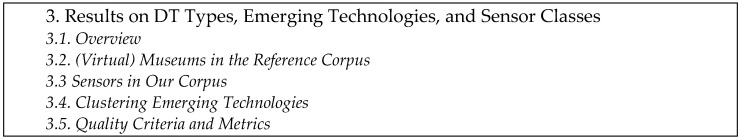
Section 3, subsections and topics.

**Table 1 sensors-23-01583-t001:** Publication data sheet for reference und corpus literature.

Parameter	Value
Title	[46] Tao, F.; Xiao, B.; Qi, Q.; Cheng, J.; Ji, P. Digital Twin in Industry: State-of-the-Art. *Journal of IEEE Transactions on Industrial Informatics* **2019**, *15(4)*, 2405–2414. https://doi.org/10.1109/Tll.2018.2873186 [CrossRef]
AA	Industry 4.0: product design, prognostics, and health management, factory of future
ET context	IoT Smart manufacturing, cyber physical systems, and data science and analytics
ET	Man-machine interaction and further examples in the history of DTs;
Tables, taxonomies	Methodology of screening paper databases: type of paper, search string and period, twin definitions and paradigms, DT classification in industry and application areas, but benefit and drawbacks if addressed;
Sensors	Sensor-based data for environmental data, to diagnose the damage size, location, and other failure information;
QC	Verification, validation and accreditation: consistency, validity, and reliability; maintenance efficiency, and accuracy;
FoDT	Communication, interaction, collaboration, service, etc.
DT	Industry: data, modeling, fusion, simulation; interaction, collaboration and service; taxonomy of DT contexts, and research areas for digital twin analytics;
TS/NR	2010–2018, 74
Future directions	Unified DT modeling method for aerospace, smart manufacturing and smart cities, construction, medical sector, robotics, ships, automobiles, rail transit, industrial engineering, agriculture, mining, energy, and environment;

**Table 2 sensors-23-01583-t002:** DT dimensions, emerging technologies and sensor classes.

DT Dimensions	Emerging Technologies	Sensor Classes
1: HMI-centric 2: HMI-centric, various 3: Content-centric, risk-informed 4: Content-centric, risk-informed 5: HMI-centric; XR device-centric 6: XR device-centric 8: HMI/content-centric 9: Content/geometry-centric 10: Content/com-centric	XR, HMD AI and ML in biomedical science Smart manufacturing, CC, AI, IoT Various ET for DT and situated analytics XR in surgery XR in Health 4.0 3D in Humanities and Education Reconstruction using photogrammetry 3D Printing 5G-TSN integration	Built-in cameras Activity recognition, remote sensing Environmental data, risk sensing Remote environmental risk sensing Measurement of kinematics, localization Glasses, gloves, trackers, helmets Scanning for point clouds and HMI Photogrammetry laser scanning, CCD IoT/CPS-based sensing, drones
11: Risk-informed 12: Complete, cognitive 13: Com/ HMI-centric 14: Geometry-centric 15: Various partial or complete 16: Com-centric/bio-informed 18: HCI-centric 19: Augmented/geometry-centric 20: Col-centric, complete	Sustainable Resource Management VR, AR immersion in metaverse VR in Education Laser scanning, photogrammetry Industry 4.0 technologies Wearables Multimodal interface technologies XR technologies in smart cities Digital participatory planning	Various bio-sensory systems Multimodal sensing, shock, eye tracker Optical sensors, gloves Capturing 3D point clouds, TLS, radar Drones, autonomous vehicles, wearable Wearables/bio-sensors for healthcare Motion tracking, inbuilt sensors IoT sensors, location, 360° camera Data to manage city’s assets and services
21: Com/HCI-centric, bio-informed 22: Com/HCI-centric 23: Com-centric 24: Asset/content-centric 25: Urban, geography-centric, risk-informed DT 26: Com/HMI-centric 27: Com-centric 28: Geometry-centric 29: Geometry-centric 30: HCI-centric DT	AR, Immersion in Hospitality Services (Blended Web mobile) learning, VR ANN in learning Laser scanning ICT technologies in Heritage, various IoT technologies AR in the smart factory environment Industry 4.0 technologies, HRC Geometric digital twinning DL, BIM, IoT, feature detection HCI for disabled people, wearables	Smartphone built-in, multimodal, haptic Sensors to support mobile technologies Multimodal sensing TLS, structured light sensors, drones Environmental, indoor position Multipurpose measurement, RFID RFID, beacons, optical, gesture Smart optical for process control, AR TLS, point clouds RGB-D sensor, LiDAR, TLS, RFID Touch, motion, proximity, light, senses
31: Com-centric, bio-informed 32: Risk-informed, bio-informed 33: HCI-centric 35: Cognitive DT 36: Com-centric 37: HMI-centric 38: Content-/geometry-centric 39: Various, risk-informed 40: Augmented 41: Com/HCI-centric 42: Various DT in Heritage 4.0 43: Robot DT	ML, information and content retrieval Law, governance technologies Nonverbal HCI Smart manufacturing DT technologies XR in smart construction Drone reconstruction, photogrammetry Web 3D, ICT, 3D modeling & printing DRM, crowdsourcing, immersion Photogrammetry, VR/AR in metaverse Interaction/immersion in virtual space BIM, VR, AR, Heritage, semantic Web IoT, AR, VR, Robots	Environmental, smart home, wearable Security issues, risk, bio, environmental Physiological modalities, contextual Sensing attributes and behaviors RFID, WSN for real-time states Ultrasonic distance, accelerometer Drone, 3D LiDAR, LOAM, 3D scanning Risk, proximity, environmental, physical properties sensors Movement detection, VR glasses Support for understanding, interaction 3D scanning, trigger sensors, humidity Localization, drone sensing, MV

**Table 3 sensors-23-01583-t003:** Clustering the corpus with respect to AA, stages in life cycle, ET, DT dimensions and features.

Application Area	Life Cycle	Technologies	Content	Communication	Collaboration
Smart cities	Design, modeling BIM, CIM, [4,25] Operating DT	Immersion [19], sensing [33], sustainable management [11]	Asset, object [4,11]	Visualization, interaction, immersion [19,25,31,33]	Participation, engagement [20]
Smart manufacturing, logistics, transportation	System modeling [3] and generating	Digital transformation, VR [1,37]	Item [10,26]	Media object [36]	Stakeholders [15,27], Decision making [35],
3D technologies in human research, Education	Design, modeling	AI, ML [2,3,32] VR [8,13]	Object [3,8,41]	Asset [3,8,13] Interaction [22,23]	Interdisciplinary collaboration [2,3,8]
XR in Health 4.0	Modeling Operating [21]	VR, XR [5,6]		Immersion [5,6]	
3D Reconstruction	Geometric modeling [28] Generating [29]	Photogrammetric, TLS [9,14,24] Drone, AI [38]	Architecture [9]		
Metaverse— immersive simulation	Designing, modeling, operating	AI, VR, AR, Robotics [12,39,40,41,42,43]	CP Social Eco-Society System [43]	Interaction, Immersion [39,41]	Stakeholders [39,41]
Disability and multi-modal mobile media, Healthcare	Designing, modeling, evaluating	Wearables (scanning) devices Mobile networks		Disability and haptic mobile media [16,30] Spatial interfaces for 3D visualization [18]	Storytelling [43]

**Table 4 sensors-23-01583-t004:** Five most important dimensions of DT in our corpus-quality criteria and metrics.

Content-Centric DT	Communication-Centric DT	Collaboration-Centric	User-Centric DT	Evolved DT
Model/process design: accuracy, performance, consistency, data quality Creation: accuracy, compliance, scalability, visual fidelity, reconstruction vs. cost optimization Operation: simulation accuracy, performance, efficiency, security Quality control during life cycle: monitoring, reliability, V&VA	Task-oriented performance Data quality: availability, correctness, completeness, integrity HCI: immersion, comfort, performance, efficiency, effectiveness, adequacy, robustness ICT: privacy, security, reliability QoS network: technical quality parameter, robustness	Level 1: Organizational Level 2: Descriptive, process-related Performance, satisfaction, effectiveness, efficiency Utility, stimulation, engagement	User experience (UX) Trust, acceptance, utility Quality of service (QoS) User engagement UE, stimulation, excitement, perception Usability, learnability, utility, attractivity, comfort	Management, mission, impact, innovation power in all fields Quantity and extent of new concepts and technologies Reputation
Accuracy: forms of [5,8,9,12,13,14] accuracy metrics [12,14,28,29] consistency, accuracy [3] Model/process: V&VA [3,11,12,28,39], data quality [14,15] model accuracy [38,42] model reliability [37] accuracy, performance [4,6] Construction: accuracy of scanning [14], reliable [9], repeatability resolution [24] of geometry-centric DT [28,29] point cloud optimization [40] performance, efficiency in content creation [28,29] efficiency and scalability, accuracy in content creation [31] accuracy and efficiency in visual fidelity [9] Operation: reliability [16] of computer security system [32] Performance metrics [16,36,37] Accuracy, performance in process/ simulation [26,27,32,35,36,38] Life cycle monitoring [36]	Task/system performance [1,2,8,13,18,23,37] Data: accuracy, privacy/security [2,26,32,37], availability, robustness [8], integrity [9], sensor robustness [10], reliability [25] Data quality [15,16,26,32,42] Model performance [23] Sensor data accuracy [37], accuracy, repeatability, and resolution [24] Immersion framework [21] Accuracy of multimodal user description/manifestations [33] ICT: reliable and secure [35], reliability, privacy, security [16] efficiency in various contexts [14,16] cost-effective visualization [19] QoS Time-Sensitive Networking (TSN) with 5G [10] HCI, Human–machine communication [43]	Performance, efficiency in interdisciplinary collaboration [2,8], in organizational practices [15] in co-working spaces [27] assessing effectiveness [20] engagement [20] Multidisciplinary team performance [41] Speedy and precise communication and collaboration of stakeholders [15] Cognitive DT: performance, efficiency in organizational and process-related quality Collaboration of connected DTs for complex processes, decision making during life cycle [35] Storytelling [43]	UX [2,4,8,13,15] UX using haptic technologies [30] trust, acceptance, utility [5], QoS [10] Usability, ease of use [16,20] Comfort [6,21] Usability: SUS [21] Attractivity: UES [21] UE [22] Excitement, level of measurement of physiological responses [25] Efficacy, usability, ease of use and usefulness [43]	Reputation [25,32] Intelligent management and reputation assessment [16] Cost-efficient management, efficiency [25,37] Technological readiness [35] Time, cost and quality management [41]

**Table 5 sensors-23-01583-t005:** Future technical, thematic and methodical work in the light of DT’s benefits and drawbacks.

Interaction and Devices	Methodical Issues	New Approaches in Applications Areas	New Insights through Application of Special Techniques or Methodologies	Benefits/Drawbacks
Technological progress in Industry 4.0 devices [1] Multiple synchronized cameras to supervise assets [9] Multimodal content representation, perception [12] VR technology in education [13] Hybrid interaction techniques for 3D visualization purposes [18] Immersion: does the sensory construct foster interaction through VR technologies? [21] Reimagining the larger project of haptic/future media, its social entailments, uses, design and policy [30] Fully immersive metaverse, integration of heritage DT in different meta verses [40] IoT devices, cardboard based HMD’s [43]	Overcome the limitations of AI&ML use in economy and society [2] Expand geographic virtual city miniature to the use of real data sets from IoT sensors [19] Shift from data to knowledge management, filtering techniques and semantic links to support the creation of the different knowledge patterns [25] Consider the impact of changeover from existing assembly to 4.0 systems [26] Building intelligent legal decision support systems [32]	Combination of remote sensing and ML [2] Identify further VR applications in cardiology [6] AEC cost control, use of AI [14] Considering alternative city planning approaches [20] Extend review parameters: time span, corpus of works analyzed; innovation in higher education, but also in middle and early education as well as educational inclusion as a new research topic [22] Further scenarios concerning the active social experience of audiovisual content in the framework of museum and cultural projects, applications in therapy and rehabilitation, [33] Development of the use of crosscutting technologies in DTs and relevant use cases [39] CPS Eco—Society System for CH [43]	Unified DT modeling method needed [3] Usage of fused DT in city planning to communicate plans to public [4] Further research on DT with HMI in surgery vs conventional methods [5] More high-quality DTs of artefacts in cities [24] Detection of potentially malicious forces to shape the future of technological systems and their impact on the population, e.g., trust [27] Need for automating the PCD—(point cloud data) to Geometry-centric DT process: object detection and model fitting—account for uncertainties in data [28] Enhance DT application opportunities; use of AI to make it cost-effective to implement [29] Visual analytics in geo-aware DT [31] Unify and align the relevant DT standards developed by different Standards Developing Organizations in the future [35] Specific research challenges: application of a DT paradigm for improving the sustainability performances in each application context; Standards and communication protocols to ensure interoperability over the whole life cycle; Provide design criteria and constraints where reference architectural aspects, information models and communication protocols are clearly defined [36] DT to improve production of future generations of vehicles to fulfill certain QC [38] During-time DT (dtDT) supports perception in VR environment [41] Refinement of DT ontology for Heritage DT [42], Robots-oriented DT [43]	More virtual visits to tourist destinations and 3D modeling from crowdsourced imagery [8] Overcome disadvantages of VR and AR technology: motion sickness, ethical concerns, and lack of privacy [11] Overcome digital disruption of the market rules [15] Overcome critical challenges from data acquiring and processing, communications, security, privacy, hardware limitations, and user adoption in wearable technology [16] Further model verification with larger data size and other ML methods, identifying personal feed back w.r.t. personalized learning requirement [23] Research limited by its being a laboratory prototype [37] The use of VR within the CH can represent an even more valuable bridge of knowl edge and understanding between the user and the built environment [41]

## Data Availability

Not applicable.

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
