# Peer review of "Digital Twins and Enabling Technologies in Museums and Cultural Heritage: An Overview"

_sensors, 2023, doi:10.3390/s23031583_

Round 1

Reviewer 1 Report

The paper is well written and would contribute to the existing body of knowledge if published. However, the following should be addressed to further improve the paper:

1. There are some grammatical and syntax issues with the current paper, a proofreader may be engaged to further improve the paper

2. From the topic, digital twins is one of the emerging technologies so combining the two in the topic looks inappropriate. 

3. This is a review paper and should be stated accordingly. 

4. The first statement of the abstract is confusing, it seems the authors just want to mention "sensor" for the paper to fit-in into the journal. There is a need to reconstruct most of the statements.

5. The first paragraph of the introduction is not coherent, there is a need to rework this and other to improve the readability of the paper.

6. The authors need to understand the difference between digitization and digitalization and use the appropriate word in the paper. 

7. What is the rationale and justification for the paper? There is a need to explain the existing problem that warrant this manuscript

8. The referencing style should be improved. the first citation is number 35 followed by 11 in stead of starting from 1 and in ascending order

9. Since the study is about museum and cultural heritage, authors should explained how the ensure that the articles considered for the study are not generic or from other industry

10. The use of abbreviation "DT" can mean digital twin or digital technology since the authors discussed the two terms. Authors should either state one in full or find a means to differentiate the two in the paper

11. There should not be citations in further works and conclusion sections.

12. The implications of the study is not clear, authors should explain the practical, theoretical and if possible methodological; implications and contributions of the study.

13. There is a need to state the limitations of the study before explaining areas for further study

14. In the conclusion, the authors made reference to questionnaires that were not mentioned earlier. the concluding part also made reference to another publication/web page, I doubt this is in order!

Author Response

Reviewer 1:

The paper is well written and would contribute to the existing body of knowledge if published. However, the following should be addressed to further improve the paper:

We thank the anonymous reviewer for the careful reading of our manuscript and many helpful and constructive comments and suggestions. Below, we respond in detail to the reviewers' comments and explain the changes in the revised submission.

  1. There are some grammatical and syntax issues with the current paper, a proofreader may be engaged to further improve the paper

We subjected the paper to professional proofreading in the USA by an expert before its submission and after correction.

  1. From the topic, digital twins is one of the emerging technologies so combining the two in the topic looks inappropriate

Thanks for pointing out this error; we have corrected this inaccuracy. In the vast majority of definitions available, the DT is introduced as a model or concept, but not as an ET.

By the way, the second chapter is devoted only to the key topics DT and ViM.

  1. This is a review paper and should be stated accordingly. 

This has been taken into account and accordingly in the changed title, new abstract and further wording.

  1. The first statement of the abstract is confusing, it seems the authors just want to mention "sensor" for the paper to fit-in into the journal. There is a need to reconstruct most of the statements.

The paper addresses sensors in the context of DTs and ViMs 121 times, mostly in citations, sensors play a very important role in DTs, which is shown in the text analysis of the 40 corpus papers, they are indispensable for communication-centric, augmented, and risk- and bio-informed DTs.  

  1. The first paragraph of the introduction is not coherent, there is a need to rework this and other to improve the readability of the paper.

The introduction was revised, the first paragraph was shortened, the second deleted, comments on the rationale added in its place, and a summary with an outlook on the other sections added at the end.

  1. The authors need to understand the difference between digitization and digitalization and use the appropriate word in the paper. 

The entire text has been checked for correct use of both terms: digitization means conversation of information/objects to be processed by computers, i.e. in our case asset, artwork and their surrounding in exhibition spaces; digitalization means using digital technologies to transform physical models and processes, e.g. path planning, navigation or to realize virtual outdoor museums.

  1. What is the rationale and justification for the paper? There is a need to explain the existing problem that warrant this manuscript

We added a paragraph in the introduction and further explanation in the conclusion.

  1. The referencing style should be improved. the first citation is number 35 followed by 11 in stead of starting from 1 and in ascending order

The paper uses two independent referencing systems: the examined corpus is listed in entries 1-43, and references to required literature in the fields examined are listed in entries I1-I43. Both are visibly separate in the references.

The latter references are introduced in their natural order, the former globally in the abstract. Their order results from the Scopus search that are displayed starting with the publication year 2022 and descending  (cf. Figure A1 1-34) and those in other databases (35-43). They are quoted continuously in the text depending on the context, not in ascending order of numbering

  1. Since the study is about museum and cultural heritage, authors should explained how the ensure that the articles considered for the study are not generic or from other industry

The study is consistent insofar as the museums found today reflect all areas of cultural life including architecture, manufacturing, etc. (cf. subsection 2.2.). In addition, comprehensive technologies are used, educational goals are pursued, and dealing with artefacts from all areas of everyday life is simulated. In this respect, starting with Architecture and Education 4.0, significant areas of industrial and cultural transformation are involved.

  1. The use of abbreviation "DT" can mean digital twin or digital technology since the authors discussed the two terms. Authors should either state one in full or find a means to differentiate the two in the paper

The term “digital technology” only appears once in the text and is not abbreviated.

  1. There should not be citations in further works and conclusion sections.

We have differentiated between “Further work” in this submission and as a topic in subsection 4.2 in the examined corpus.  Citations are only included for the latter.

  1. The implications of the study is not clear, authors should explain the practical, theoretical and if possible methodological; implications and contributions of the study.

We added a sub-section 4.3 Implications in section 4.

  1. There is a need to state the limitations of the study before explaining areas for further study

We added a corresponding paragraph in the conclusions.

  1. In the conclusion, the authors made reference to questionnaires that were not mentioned earlier. the concluding part also made reference to another publication/web page, I doubt this is in order!

This issue was corrected, the paragraph has been moved to a more appropriate place.

Reviewer 2 Report

The Introduction is confusing, and it needs to be better organized. The Introduction should provide a brief contextualization of the research and clearly present the objectives at the end. In the current file, the Introduction contains little theoretical basis (practically no references) and still brings theoretical reference and methodology elements.

The Materials and Methods section also contains elements out of order. The Materials and Methods section should contain only the research steps, not the results. The whole section should be reorganized

It is a very interesting article because data digitalization and virtualization are gaining a lot of prominences worldwide because these technologies bring the possibility of simulation of environments, processes, equipment, among others. It needs a little more work in organizing the sections. Congratulations, and I wish you a great job on the corrections.

Author Response

Reviewer 2:

We thank the anonymous reviewer for the careful reading of our manuscript and many helpful and constructive comments and suggestions. Below, we respond in detail to the reviewers' comments and explain the changes in the revised submission.

The Introduction is confusing, and it needs to be better organized. The Introduction should provide a brief contextualization of the research and clearly present the objectives at the end. In the current file, the Introduction contains little theoretical basis (practically no references) and still brings theoretical reference and methodology elements.

We gave the paper a new title. The abstract was rewritten and concluded with a clear  recommendation. The introduction was revised, the first paragraph was shortened, the second deleted, comments on the rationale added in its place, and a summary with an outlook on the other sections added at the end.

The paper uses two independent referencing systems: the examined corpus is listed in entries 1-43, and references to required literature in the fields examined are listed in entries I1-I43. Both are visibly separate in the references.

The latter references are introduced in their natural order, the former globally in the abstract. Their order results from the Scopus search that are displayed starting with the publication year 2022 and descending  (cf. Figure A1 1-34) and those in other databases (35-43). They are quoted continuously in the text depending on the context, not in ascending order of numbering, [1-43] and ascending [I1-I9] in the introduction and further on.  We added the corpus references dealing with various types of DTs at the end of section 1.

The Materials and Methods section also contains elements out of order. The Materials and Methods section should contain only the research steps, not the results. The whole section should be reorganized

Section 2 has been better structured and the presentation revised.

Section 2 refers to basic concepts of DTs and ViMs as well as methodological steps to investigate PTs and DTs in the context of application areas and required technologies.

The section Materials and Methods lays out the concepts and properties of DTs and ViMs in two subsections and subdivisions from different perspectives to enable the examination of the mapping Reference ↔ DT Dimensions ↔ Emerging Technologies ↔ Sensor Classes. This screening of the reference corpus is summarized in Table 2 and provides a starting point for the presentation of our results in Section 3, allowing us to relate the occurrence of digital twins by type and features to the enabling technologies and sensors and to derive conspicuous correlations and their clusters.

Fundamental to this approach is the introduction of dimensions of DT starting from the three main types--content-, communication-, and collaboration-centric DT (ViM)--and their further or overarching features. These facts and further concepts are provided in Section 2 before conclusions can be drawn in Section 3 with its subsections and the different clusterings of the corpus literature derived w.r.t. application areas, stages in life cycle, ET, DT dimensions and features.

By the way, we added a sub-section 4.3 Implications in section 4 and stated the limitations of the study in a corresponding paragraph in the conclusions.

It is a very interesting article because data digitalization and virtualization are gaining a lot of prominences worldwide because these technologies bring the possibility of simulation of

Reviewer 3 Report

1. The title should be specific to the manuscript.

2. The problem should be in the abstract with a static conclusion.

3. The references in the introduction are not in ascending order, and every paragraph needs a reference.

4. The problem must be very clearly and specific

5. Table 1, should write in a paragraph or some keywords with reference not only references

6. Table 2. References needed and application

7. Table 3. must be in a separate file and not in the middle of the manuscript.

  1. Method and materials should be outlined in a diagram..

9. Figure 1 is used as a reference. If it is not yours, please remove it or make your own figure.

10. It's a review paper, so Table 4 must have references in each dimension

11. similar table 5 needs keywords in each of the columns not only reference

12. This is very significant to put the applicatiobefore futurere work

Author Response

Reviewer 3:

We thank the anonymous reviewer for the careful reading of our manuscript and many helpful and purposeful comments and suggestions. Below we respond in detail to the reviewers' comments and explain the changes in the revised submission.

  1. The title should be specific to the manuscript.

We gave the paper a new title.

  1. The problem should be in the abstract with a static conclusion.

The abstract was rewritten and concluded with a clear recommendation.

  1. The references in the introduction are not in ascending order, and every paragraph needs a reference.

The paper uses two independent referencing systems: the examined corpus is listed in entries 1-43, and references to required literature in the fields examined are listed in entries I1-I43. Both are visibly separate in the references.

The latter references are introduced in their natural order, the former globally in the abstract. Their order results from the Scopus search that are displayed starting with the publication year 2022 and descending  (cf. Figure A1 1-34) and those in other databases (35-43). They are quoted continuously in the text depending on the context, not in ascending order of numbering.

  1. The problem must be very clearly and specific

The abstract was changed and the introduction thoroughly revised, a rationale added in subsection 1.1..

  1. Table 1, should write in a paragraph or some keywords with reference not only references

Table 1 was replaced by a more appropriate figure.

  1. Table 2. References needed and application

The revised text explains that Table 2 is a representation of an index card format developed for the study to examine the relationships between the key terms.

  1. Table 3. must be in a separate file and not in the middle of the manuscript.

 Table 3 has been moved to the newly added Appendix A. 

  1. Method and materials should be outlined in a diagram.

A corresponding figure has been inserted at the beginning.

  1. Figure 1 is used as a reference. If it is not yours, please remove it or make your own figure.

The figure is based on an earlier version that the authors used in their publication I30 and supplemented with significant new aspects.

  1. It's a review paper, so Table 4 must have references in each dimension

The links were added, each row of the table 2 (2 instead of 4) concerns a unique corpus paper.

  1. similar table 5 needs keywords in each of the columns not only reference

The keywords were added.

  1. This is very significant to put the application before future work

We have differentiated between “Further work” in this submission and as a topic in subsection 4.2 of the discussion in the examined corpus.  Citations are only included for the latter.

Round 2

Reviewer 1 Report

The paper has been improved based on the earlier comments

Author Response

Reviewer 1:

We thank the anonymous reviewer for the careful reading of our manuscript and many helpful and constructive comments and suggestions. Below, we respond in detail to the reviewer's comments and explain the changes in the second revised submission sensors-2161732_rev_rev_26_01_2023.

The paper has been improved based on the earlier comments

We thank the reviewer for all the comments in the two reviews. The comments have helped to improve the quality and comprehensibility of the text.

 (x) Extensive editing of English language and style required

We checked the text thoroughly for stylistic flaws, formatting and spelling errors as well as the correctness of the links. 

All submitted versions underwent a thorough language revision in three rounds by a professional native speaker from USA.

Ms. Elizabeth von Schoff (https://www.the-efa.org/memberinfo/e-von-schoff-11636/) has been correcting all my scientific publications in journals and proceedings for the past 25 years without any criticism from reviewers or organizers. She was employed for many years at the University of Duisburg-Essen as an English language lecturer, but returned to California about 20 years ago and is doing freelance work for publishers and educational institutions.

For your information, besides the second revision of the paper and a comparison of the current with the previous version, we attach the following document

sensors-27161732_correction_history_E_J_von_Schoff:

Corrections by Ms. Elizabeth von Schoff dated 29.12.2022, 09.01.2023, 25.01.2023

Invoices from Mrs. Elizabeth von Schoff

Please see the two attachments Corrections and changes

Reviewer 2 Report

Thanks for answer all my appointments.
Only section 1.2 is repeated, it should be section 1.3.

Author Response

Reviewer 2:

We thank the anonymous reviewer for the careful reading of our manuscript and many helpful and constructive comments and suggestions. Below, we respond in detail to the reviewer's comments and explain the changes in the second revised submission sensors-2161732_rev_rev_26_01_2023.

Thanks for answer all my appointments.

We thank the reviewer for all comments in the two reviews. The comments have helped to improve the quality and comprehensibility of the text.

Only section 1.2 is repeated, it should be section 1.3.

The incorrect numbering of subsection 1.3 in section 1 is now corrected.

(x) English language and style are fine/minor spell check required

We checked the text thoroughly for formatting and spelling errors and correctness of the links. 

By the way and for your information: all submitted versions underwent a thorough language revision in three rounds by a professional native speaker from USA.

Ms. Elizabeth von Schoff (https://www.the-efa.org/memberinfo/e-von-schoff-11636/) has been correcting all my scientific publications in journals and proceedings for the past 25 years without any criticism from reviewers or organizers. She was employed for many years at the University of Duisburg-Essen as an English language lecturer, but returned to California about 20 years ago and is doing freelance work for publishers and educational institutions.

Besides the second revision of the paper and a comparison of the current with the previous version, we attach the following document

sensors-27161732_correction_history_E_J_von_Schoff:

Corrections by Ms. Elizabeth von Schoff dated 29.12.2022, 09.01.2023, 25.01.2023

Invoices from Mrs. Elizabeth von Schoff

Please see the 2 attachments changes in the first revision and correction history

Reviewer 3 Report

The authors addressed all my previous comments

Author Response

Reviewer 3:

We thank the anonymous reviewer for the careful reading of our manuscript and many helpful and purposeful comments and suggestions. Below we respond in detail to the reviewer’s comments and explain the changes in the revised submission sensors-2161732_rev_rev_26_01_2023.

(x) English language and style are fine/minor spell check required

We checked the text thoroughly for formatting and spelling errors and correctness of the links. 

By the way and for your information: all submitted versions underwent a thorough language revision in three rounds by a professional native speaker from USA.

Ms. Elizabeth von Schoff (https://www.the-efa.org/memberinfo/e-von-schoff-11636/) has been correcting all my scientific publications in journals and proceedings for the past 25 years without any criticism from reviewers or organizers. She was employed for many years at the University of Duisburg-Essen as an English language lecturer, but returned to California about 20 years ago and is doing freelance work for publishers and educational institutions.

Besides the second revision of the paper and a comparison of the current with the previous version, we attach the following document

sensors-27161732_correction_history_E_J_von_Schoff:

Corrections by Ms. Elizabeth von Schoff dated 29.12.2022, 09.01.2023, 25.01.2023

Invoices from Mrs. Elizabeth von Schoff

The authors addressed all my previous comments

We thank the reviewer for all the comments in the two reviews. The comments have helped to improve the quality and comprehensibility of the text.

Please see the two attachments

Changes in the revised version and correction history
